# Influence of Phosphodiesterase Inhibition on CRE- and EGR1-Dependent Transcription in a Mouse Hippocampal Cell Line

**DOI:** 10.3390/ijms21228658

**Published:** 2020-11-17

**Authors:** Erik Maronde

**Affiliations:** Institute for Anatomy II, Faculty of Medicine, Goethe-University, Theodor-Stern-Kai-7, 60590 Frankfurt, Germany; e.maronde@em.uni-frankfurt.de

**Keywords:** hippocampus, PDE inhibition, CRE-dependent transcription, EGR1-dependent transcription

## Abstract

Signaling pathways, depending on the second messenger molecule cAMP, modulate hippocampal cell signaling via influencing transcription factors like cAMP-regulated element-binding protein (CREB) or early growth response 1 EGR1/Krox24/zif268/ZENK (EGR1). Here, we investigated two reporter cell lines derived from an immortalized hippocampal neuronal cell line stably expressing a CRE- or EGR1-luciferase reporter gene (HT22CREluc and HT22EGR1luc, respectively). The cells were subjected to phosphodiesterase inhibitors and other cAMP-modulating agents to investigate dose- and time-dependent phosphodiesterase (PDE)-mediated fine-tuning of cAMP-dependent transcriptional signaling. The non-isoform-specific cyclic nucleotide phosphodiesterase (PDE) inhibitor isobutyl-methyl-xanthine (IBMX), as well as selective inhibitors of PDE3 (milrinone) and PDE4 (rolipram), were tested for their ability to elevate CRE- and EGR1-luciferase activity. Pharmacological parameters like onset of activity, maximum activity, and offset of activity were determined. In summary, phosphodiesterase inhibition appeared similarly potent in comparison to adenylate cyclase stimulation or direct activation of protein kinase A (PKA) via specific cAMP agonists and was at least partly mediated by PKA as shown by the selective PKA inhibitor *Rp*-8-Br-cAMPS. Moreover, transcriptional activation by PDE inhibition was also influenced by organic anion-exchanger action and interacted with fibroblast growth factor (FGF) receptor-mediated pathways.

## 1. Introduction

Intracellular levels of cAMP are regulated by substances activating or inhibiting the enzyme adenylate cyclase (AC), which catalyzes the reaction of ATP to cAMP [1]. Elevated levels of cAMP lead to binding to various effector proteins like cAMP-dependent protein kinase A (PKA), exchange factor directly activated by cAMP (EPAC), or different kinds of cyclic nucleotide-gated (CNG) ion channels [2,3,4,5]. 

The degradation of cAMP is accomplished by the enzyme cyclic nucleotide phosphodiesterase (PDE), of which eleven isozyme families have been described so far [6]. Through the regulation of cAMP levels, PDEs are involved in synaptic plasticity processes such as those observed during memory processes, aging, and neurodegeneration [6,7,8]. 

The hippocampus plays a central role in the storage and retrieval of memory [9,10,11]. Transcriptional processes regulated by the second messenger 3’,5’-adenosine monophosphate (cAMP) are important in these processes [11]. Moreover, an influence of PDE inhibition on hippocampal neuronal signaling is of potential clinical importance since some isoform-specific inhibitors already serve as medications for cardiovascular and other systemic diseases [12]. The upregulation of PDE4 during sleep deprivation has been described to lead to memory deficits, which were compensated by application of the PDE4-specific inhibitor rolipram [13]. However, whether inhibition of PDE is sufficient for the activation of CRE- and EGR1-dependent transcriptional pathways similar to AC activation is not clear and will be investigated here. 

For a closer pharmacological characterization of the impact of PDE inhibition on cAMP-dependent transcriptional signaling in mouse hippocampal neuronal cells, we constructed two immortalized reporter cell lines [14], stably transfected with either a cAMP-regulated element (CRE) or an “early growth response 1” (EGR1) luciferase reporter plasmid [14,15]. We used these cell lines (HT22CREluc and HT22EGR1luc) for the investigation of dose- and time-dependency of the response toward different PDE inhibitors. These reporter gene cell lines are suitable for investigating with narrow (minutes) temporal resolution of the transcriptional response. They also provide temporal comparison of onset of response, maximum response, and offset of response to the substances applied. Much more than single-point, single-dose experiments, this method provides the opportunity to evaluate relative potency/effectivity of different substances using similar (or different) signaling pathways and their interactions.

The relative potency/effectivity of action of PDE inhibitors was evaluated here by comparing their activity with that of the AC activator forskolin [16] and the PKA-specific cAMP agonist *Sp*-5,6-DCl-cBIMPS [17,18,19,20]. To evaluate the role of PKA for the transcriptional response, the specific cAMP antagonist *Rp*-8-Br-cAMPS [19,21,22] was used. Since PKA is not the only effector protein modulated by cAMP, we compared the data on PDE inhibitors, forskolin, and *Sp*-5,6-DCl-cBIMPS with those of other potential cAMP effector proteins, namely cyclic nucleotide-gated (CNG) ion channels [3,23,24] and exchange factor directly activated by cAMP (EPAC) [4,25]. 

An often-neglected factor regulating intracellular cAMP levels is the export of organic ionic substances like cAMP through the plasma membrane of the cell by organic anion exchangers (OAEs) [26]. The influence of OAEs was tested here using the OAE inhibitor probenecid [27,28]. 

Finally, the interaction between PDE and the fibroblast growth factor 1 (FGF1)- and FGF receptor-mediated and tyrosine kinase-dependent signal transduction pathway was investigated because cAMP-regulated processes are known to be able to interact with protein tyrosine kinase signaling systems [29,30].

## 2. Results 

### 2.1. Experimental Data

#### 2.1.1. Influence of Cell Number and Concentration of Luciferin on CRE- and EGR1-Dependent Luciferase Activity

For the estimation of the influence of cell number on HT22CREluc or HT22EGR1luc activity, 1000, 2500, 5000, 12,500, 25,000, 50,000, or 100,000 cells/well of a 96-well plate were seeded (plated out) and treated the next day with the indicated reagents. Control cells were not treated with the reagents. An optimal stimulant to control the level in both cell lines was reached at 25,000 cells plated per well (Figure 1A). Figure 1B shows the time-resolved (15 min bouts starting at 15 min after application) graph for 25,000 cells/plate treated with IBMX (300 µM) in comparison to control (untreated) levels (mean ± SD; *N* = 4). Note that the onset of activity and the maximum activity in the HT22EGR1luc cells (90 min/8.5 h) was later than that in the HT22CREluc cells (60 min/6 h) in the IBMX-treated cultures.

The strength of the signal also depended on the dose of luciferin in the medium. A significant difference between control and forskolin-treated cells was observed at or above 25 µM luciferin in the medium (*p* < 0.001). A stable (more than five-fold) elevation of treated cells over control values occurred for 50–500 µM luciferin in the medium (Appendix A). 

#### 2.1.2. Influence of Time and Dose of PDE Inhibitors and Other Agents on Luciferin-Dependent CRE- and EGR1-Luciferase Activity

HT22CREluc activity was time- and dose-dependently elevated by IBMX application (Figure 2). HT22EGR1luc activity was also time- and dose-dependently elevated by IBMX, which to our knowledge was shown here for the first time (Figure 2).

#### 2.1.3. Influence of Protein Kinase A (PKA) Inhibition on HT22CRE- and EGR1-Luciferase Activity Elevated by PDE Inhibitors

In HT22CREluc cells, IBMX-, forskolin-, and *Sp*-5,6-DCl-cBIMPS-induced activities were reduced by 42%, 38%, and 77% in the presence of *Rp*-8Br-cAMPS (500 µM; *p* < 0.0001), respectively, whereas activity under *Rp*-8Br-cAMPS application alone was not significantly lower than that of control (Figure 3A). In HT22EGR1luc cells, IBMX-, forskolin-, and *Sp*-5,6-DCl-cBIMPS-induced activities were 45%, 52%, and 80% lower in the presence of *Rp*-8Br-cAMPS (500 µM; *p* < 0.0001), respectively, whereas activity under *Rp*-8Br-cAMPS application alone was not significantly lower than that of control (Figure 3B). 

It has to be noted that *Rp*-8-Br-cAMPS itself has been shown to inhibit PDEs [31] in a way that was not further being investigated here.

The time course graphs showing IBMX and IBMX plus *Rp*-8Br-cAMPS revealed that, besides the reduction of the maximal activity, *Rp*-8Br-cAMPS application also delayed the onset of activity in both HT22CRE- and HT22EGR1 cells (Figure 3C,D). Note that, besides the reduction of the maximum activity, *Rp*-8Br-cAMPS application also delayed the onset of activity. It should be noted that these inhibition experiments were executed using high, effect-saturating concentrations of the activators. It can be expected that, at lower doses, the inhibition by *Rp*-8Br-cAMPS co-application will be even higher. The effect of 100 µM IBMX, for example, was reduced by 80% by *Rp*-8Br-cAMPS (500 µM), similar to what was previously shown for forskolin (1 µM) in HT22CREluc cells [14]. Here, we chose effect-saturating concentrations of IBMX, forskolin, and *Sp*-5,6-DCl-cBIMPS to compare the different PKA-activating agents and principles for the purpose of direct comparison.

#### 2.1.4. Influence of the PDE3- and PDE4-Specific Inhibitors Milrinone and Rolipram

Both the PDE4-selective inhibitor rolipram and the PDE3-selective inhibitor milrinone dose-dependently elevated HT22CREluc activity (Figure 4A). Rolipram appeared similar to IBMX in maximal response, but was, in contrast to IBMX, already active at sub-micromolar concentrations, whereas milrinone was at least two orders of magnitude less efficient. 

Besides the different dose-dependency and efficiency, rolipram and milrinone also differed in the onset of activity, maximum activity, and offset of activity (Figure 4B).

#### 2.1.5. The Influence of Organic Anion Exchangers/Transporters (OAEs)

Besides degradation through PDE and AC activation, intracellular levels of cyclic nucleotides can be regulated by export via organic anion exchangers (OAEs). We tested this influence using the broad-range OAE inhibitor probenecid (Figure 5). 

The OAE inhibitor probenecid at 100 µM significantly elevated HT22CREluc activity when applied alone (compared with control; Tukey’s multiple comparison test; *p* < 0.001) and led to further elevation of the maximal activity when combined with rolipram (Figure 5A), suggesting an additive mode of action. However, rolipram (100 µM), with the addition of probenecid (100 µM), appeared to be more than additive (synergistic) compared with rolipram alone (*p* < 0.01; ANOVA with Tukey’s post-test; Figure 5B).

#### 2.1.6. Interaction of Rolipram with the FGF Receptor/Protein Tyrosine Kinase Pathway

We finally investigated the interaction of rolipram with FGF1-stimulated FGF receptor tyrosine phosphorylation in order to evaluate the interaction of the PDE/cAMP/PKA/CREB pathway with another prominent cell signaling pathway involved in hippocampal function. We tested this influence using the physiological FGF receptor agonist FGF1 (Figure 6). 

Like probenecid (Figure 5), FGF1 also significantly elevated HT22CREluc activity when applied alone (compared with control; Tukey’s multiple comparison test; *p* < 0.01) and led to further elevation of the maximal activity when combined with rolipram (Figure 6A), suggesting an additive mode of action similar to rolipram in combination with probenecid shown above. However, with rolipram (100 µM), the addition of FGF1 (10g/mL) was not significantly additive (Figure 6B).

## 3. Discussion

The present work characterized CRE- and EGR1-dependent transcriptional responses of two reporter cell lines, derived from the immortalized mouse hippocampal neuronal cell line HT22 [25,26], to PDE inhibition and other modes of cAMP elevation and modulation. Both HT22CRE- and HT22EGR1-dependent activities displayed a robust dose- and time-dependent response to PDE inhibitors like IBMX and rolipram, the adenylate cyclase activator forskolin [4,27], and the highly PKA-specific activator *Sp*-5,6-DCl-cBIMPS [16,28,29].

We also provided evidence that a significant proportion of the PDE inhibitor-induced HT22CREluc and EGR1luc elevation was mediated by PKA activation. *Rp*-8-Br-cAMPS strongly reduces PKA activity [18,28,30,31] *Rp*-8-Br-cAMPS was shown here to strongly attenuate IBMX-elevated HT22CREluc and HT22EGR1luc activity, suggesting PKA to be at least necessary, if not sufficient, as the mediator of IBMX-induced CRE- and EGR1-mediated transcriptional responses.

The isoform-specific PDE inhibitors rolipram (PDE4) and milrinone (PDE3) displayed different HT22CRE activity profiles in term of dose-dependency, time to reach maximal response, and time to return to basal levels. Whereas rolipram appeared equally maximally effective to IBMX in HT22CREluc activity enhancement, it was active already at sub-micromolar concentrations. In the dose-dependency experiments, rolipram showed an early plateau between 1 and 100 µM and then increased further, similar to IBMX. It was tempting to speculate that the first plateau was due to specific PDE4 inhibition, whereas the second rise was due to recruitment of non-specific targets like other PDEs. These findings suggest a prominent role of PDE4 in HT22 cells as previously shown for primary hippocampal neuronal cell and slice preparations and in mice and humans in vivo [13].

To our knowledge, it was also observed here for the first time that inhibition of OAE by probenecid not only further elevated rolipram-induced HT22CREluc activity, but was also active, when applied alone. The intracellular concentration of cAMP is determined not only by the synthesis rate of adenylate cyclase, which includes the availability of cytosolic ATP and the degradation of cAMP by PDE, but also, additionally by the export of cAMP by passive membrane transport systems of which the most prominent ones are OAEs. A very minor proportion of intracellular cAMP is bound to the various cAMP-binding proteins and is usually so small that it can be neglected [8] Since probenecid only inhibits the export of cAMP through OAE, but does not change either the synthesis or degradation, it may be speculated that basal cAMP synthesis in HT22 cells is high.

Rolipram-induced HT22CREluc activity was modulated by FGF1 application. FGF1 has been shown to activate and modulate several signal transduction pathways in HT22 cells like the mitogen activated protein MAP kinases or protein kinase B [24] Here, we presented an enhancement of rolipram-induced HT22CREluc activity and a small, but significant effect, of FGF1 alone. Although not significantly synergistic like the effect of probenecid on rolipram (see above), the curve changes displayed by rolipram in combination with FGF1 were similar. It may thus be possible that FGF1 influences the OAE, but given the multitude of actions FGF1-dependent pathways influence, this is most likely just correlation, not causation.

As mentioned before, differences in onset and maximum activity were found for different PDE inhibitors, for each inhibitor in different cell lines, and also for different doses. Considering single time-point experiments, these differences may influence results. The last point had already been described before in HT22CREluc cells [14] and was confirmed here for HT22EGR1 cells and PDE inhibitors. This may be relevant for some types of experiments like those presented here. Late onset of action, for example, may lead to the erroneous conclusion that a substance is inactive, but in fact the activity has just not begun.

Both HT22CREluc and HT22EGR1luc activities were shown to be elevated by AC/cAMP/PKA-activating agents as well as PDE and OAE inhibitors. The HT22CREluc reporter line was more sensitive to the stimuli and reacted earlier that EGR1luc. There are many possible explanations for this, of which these appear to be the most likely ones: (a) the influence of cAMP elevation and PKA activation is weaker on EGR1 expression and transcriptional activity, (b) CRE activation precedes EGR1 activation, and (c) other PKA targets are necessary for EGR1 activity elevation, such as CREB-binding protein (CBP) phosphorylation [32].

In summary, we provided evidence for PDE inhibition-mediated elevated transcription via both CRE and EGR1 pathways. PDE4 appeared to be the most sensitive target PDE, but other PDEs may contribute too. The inhibition of OAE and activation of the FGF1 signaling pathway further elevated this transcriptional response, even synergistically under some conditions. Since many PDE inhibitors are in clinical use and the development of PDE-mediated elevated transcriptional activity may influence memory function, changes in memory functions have to be considered as potential side-effects under these treatments.

## 4. Materials and Methods 

### 4.1. Cell Culture

HT22CREluc cells were produced as described [14,32,33] In short, HT22EGR1luc cells were made following the same procedure described for HT22CRE, but using an EGR1 reporter plasmid kindly provided by Dr. Pavel Krejici [15] Cells were cultivated in Dulbecco’s modified Eagle’s medium (DMEM; Sigma, Deisenhofen, Germany) with 10% fetal bovine serum (FBS), penicillin/streptomycin (100 U/mL) and GlutaMax (Life Technologies, Darmstadt, Germany). Clones showing stable induction of CREluc or EGR1luc activity (more than 4-fold elevated RLU after application to 1 µM forskolin in comparison to untreated culture) were expanded and frozen at a density of 1.5 million cells/mL/vial at −80 °C in a freezing medium (ibidi GmbH, Planegg, Germany). Living cells were passaged once a week at a density of 0.1–0.5 million cells per 75 cm^2^ flask.

### 4.2. Determination of Luminescence Activity

If not indicated otherwise, 25,000 cells per well of a 96-well plate were plated out in 100 µL volume and left in the incubator overnight in order to promote adhesion. Experiments were performed in 200 µL medium/well containing a final concentration of 0.5 mM (500 µM) luciferin (if not indicated otherwise) and measured in a luminometer (BMG Lumistar; BMG, Ortenberg, Germany or Berthold Centro LB960, Berthold Technologies, Bad Wildbad, Germany) at 35 °C. Luminescence was measured for 0.1 sec per well. Luminescence data are displayed as relative luminescence units (RLU). Cells were measured after different times of incubation (usually every 15 min for up to 24 h if not indicated else) with and without chemical agents.

### 4.3. Mitochondrial Lactate Dehydrogenase Activity (Water Soluble Tetrazolium (WST-1) Assay)

After completion of luciferase experiments, cell culture medium was discarded and fresh, serum-free medium containing WST-1 was added for one hour for estimation of relative viability. The mitochondrial lactate dehydrogenase activity estimated in living cells with this test may reflect different cell viabilities, cell numbers, or a mixture of both parameters. It has to be taken into account that chemicals applied may be toxic for cells and this could be misjudged as an inhibitory effect, but may in fact only reflect toxicity. Therefore, integrity of the cells and the cell layer was verified by microscopical examination and only experiments displaying no reduced or very different viabilities between the compared treatments were shown.

### 4.4. Chemicals

Isobutyl-methyl-xanthine (IBMX), forskolin [16], probenecid [34,35], milrinone [36], and rolipram [37] (Sigma, Deisenhofen, Germany) were dissolved in dimethyl sulfoxide (DMSO). *Rp*-8-Br-cAMPS [21,22], *Sp*-5,6-DCl-cBIMPS [17,19,20], and EPAC activator [25] (BioLog Life Science Institute, Bremen, Germany) were dissolved in water. FGF1 [30] (Sigma, Deisenhofen, Germany) was stored in H_2_O/50% glycerin at −20 °C and dissolved in water or cell culture medium for the experiments. Luciferin [14] (Promega, Heidelberg, Germany) was dissolved directly in the cell culture medium at the indicated concentrations. Reagents or appropriate vehicles were applied to the media for the indicated periods.

### 4.5. Statistical Methods

If not indicated otherwise, group comparisons were made using ANOVA with Sidak’s multiple comparison as implemented in GraphPad PRISM 8.4.3 (GraphPad Software Inc., San Diego, CA, USA). Sidak’s test was preferred over Bonferroni because of its higher statistical power.

## Figures and Tables

**Figure 1 ijms-21-08658-f001:**
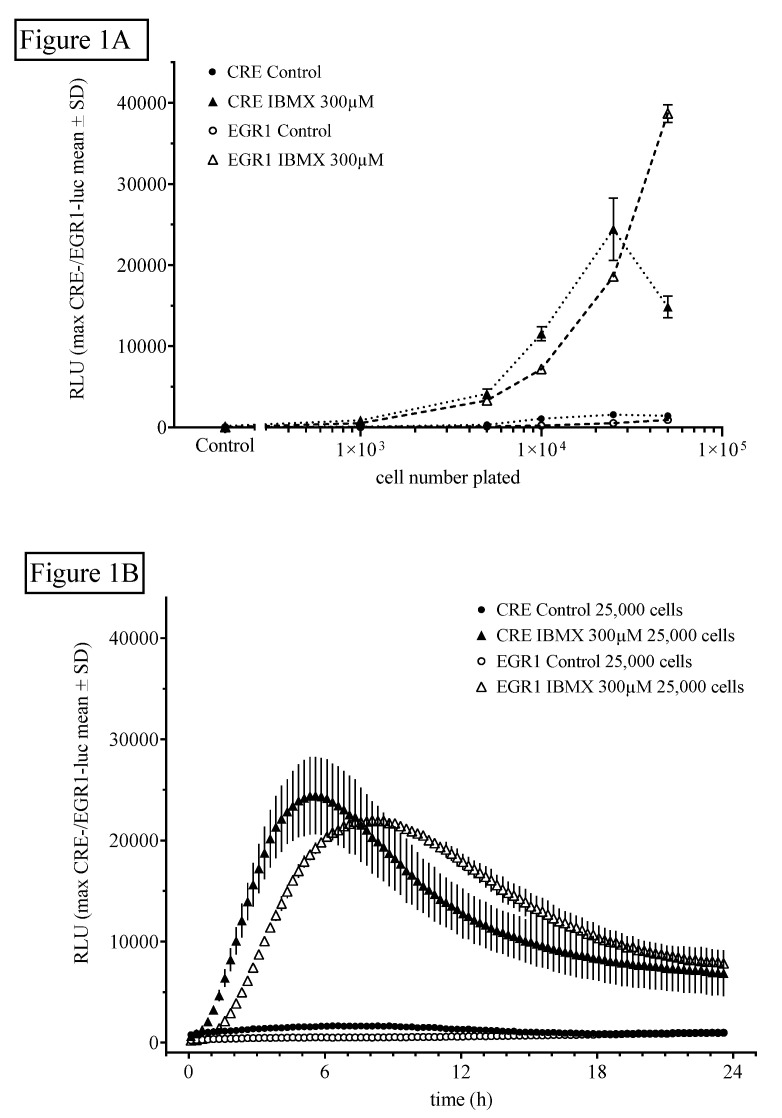
The non-selective cyclic nucleotide phosphodiesterase (PDE) inhibitor isobutyl-methyl-xanthine (IBMX) induces cyclic AMP response element (CRE)- and early growth response 1 (EGR1)-luciferase activity. (**A**) HT22CRE-luciferase (closed symbols) or EGR1-luciferase (open symbols) activity induced by the application of IBMX (300 µM) compared with untreated controls. HT22CREluc reached maximum activity at 25,000 cells, whereas HT22EGR1luc activity further increased at 50,000 cells without a sign of a plateau. Mean values with standard deviation (mean ± SD) of four repetitions (*N* = 4) measured in the Centro luminometer are shown. (**B**) Time-resolved (15 min bouts starting at 15 min after application) graph for 25,000 cells/plate treated with IBMX (300 µM) in comparison to control (untreated) levels (mean ± SD; *N* = 4). Note that the onset of activity and the maximum activity in the HT22EGR1luc cells (90 min/8.5 h) were later than that in the HT22CREluc cells (60 min/6 h) in the IBMX-treated cultures.

**Figure 2 ijms-21-08658-f002:**
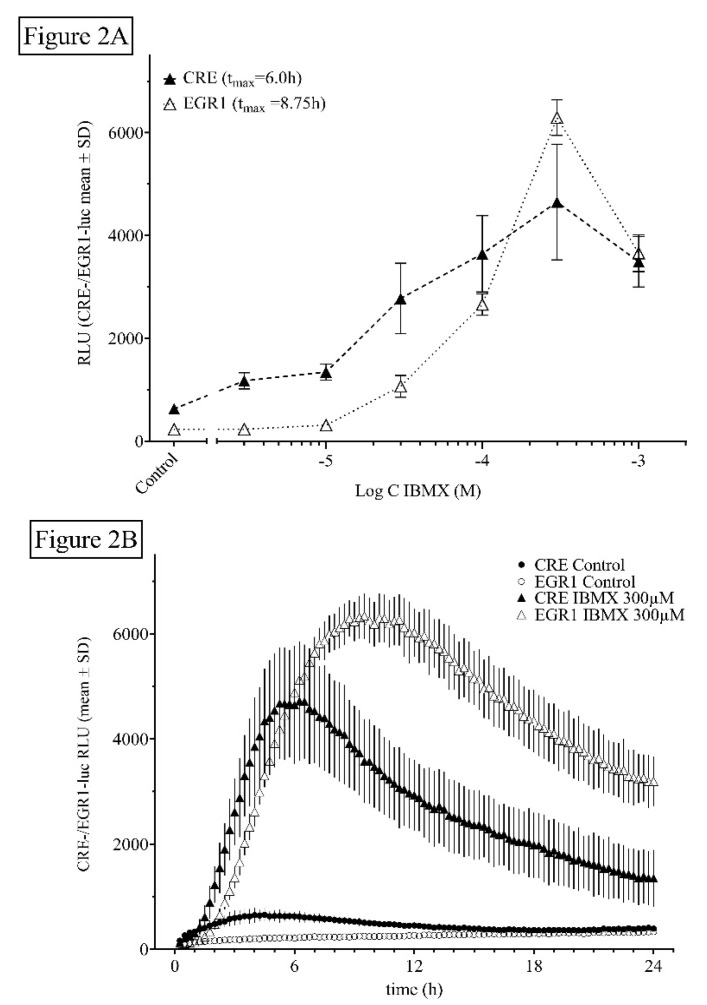
Dose- and time-dependent activity of the non-selective PDE inhibitor IBMX. (**A**) Dose-dependency of HT22CREluc and HT22EGR1luc cells (25,000 cells/well) for IBMX. Mean values with standard deviation (mean *±* SD) of four repetitions (*N* = 4) measured in the Lumistar luminometer are shown. (**B**) Time-resolved (15 min bouts starting at 15 min after application) graph for 25,000 cells/plate treated with increasing doses of IBMX in comparison to control (untreated) levels (mean ± SD; *N* = 4). Note that the onset of activity and the maximum activity in the HT22EGR1luc cells (90 min/8.5 h) were later than that in the HT22CREluc cells (60 min/6 h) in the IBMX-treated cultures. Note that the data in Figure 2B are very similar to those in 1B just with a lower relative luminescence unit (RLU) output.

**Figure 3 ijms-21-08658-f003:**
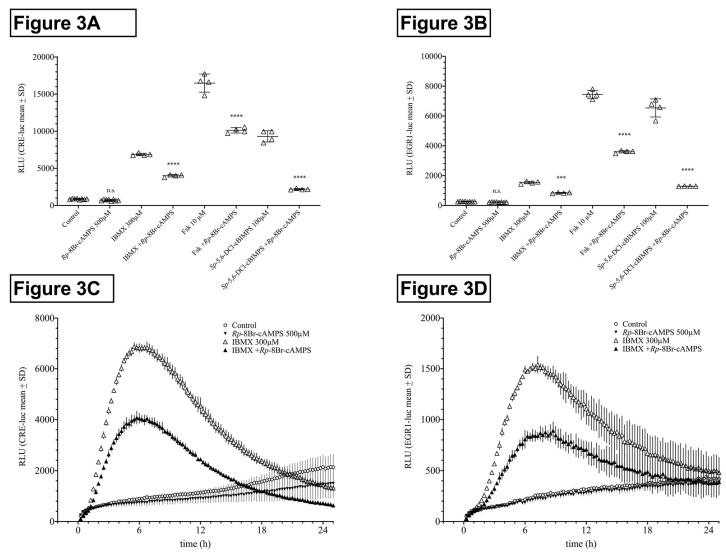
HT22CREluc and HT22EGR1luc activity of IBMX is mediated by protein kinase A (PKA). Reduction in HT22CREluc (**A**,**C**) and HT22EGR1luc (**B**,**D**) activity under inhibition of PKA activity due to pre- and co-application of the selective PKA inhibitor *Rp*-8Br-cAMPS (1 mM during pre-application and 0.5 mM during co-application). In HT22CREluc cells, IBMX-, forskolin-, and *Sp*-5,6-DCl-cBIMPS-induced activities were 42%, 38%, and 77% lower in the presence of *Rp*-8Br-cAMPS (*p* < 0.0001), respectively, whereas activity under *R*p-8Br-cAMPS application alone was not significantly lower than that of control. In HT22EGR1luc cells, IBMX-, forskolin-, and *Sp*-5,6-DCl-cBIMPS-induced activities were 45%, 52%, and 80% lower in the presence of *Rp*-8Br-cAMPS (*p* < 0.0001), respectively, whereas activity under *Rp*-8Br-cAMPS application alone was not significantly lower than that of control (mean ± SD; *N* = 4–8). Time courses of HT22CREluc (C) and HT22EGR1luc (D) activity for IBMX and IBMX plus *Rp*-8Br-cAMPS. All data are from the Centro luminometer.

**Figure 4 ijms-21-08658-f004:**
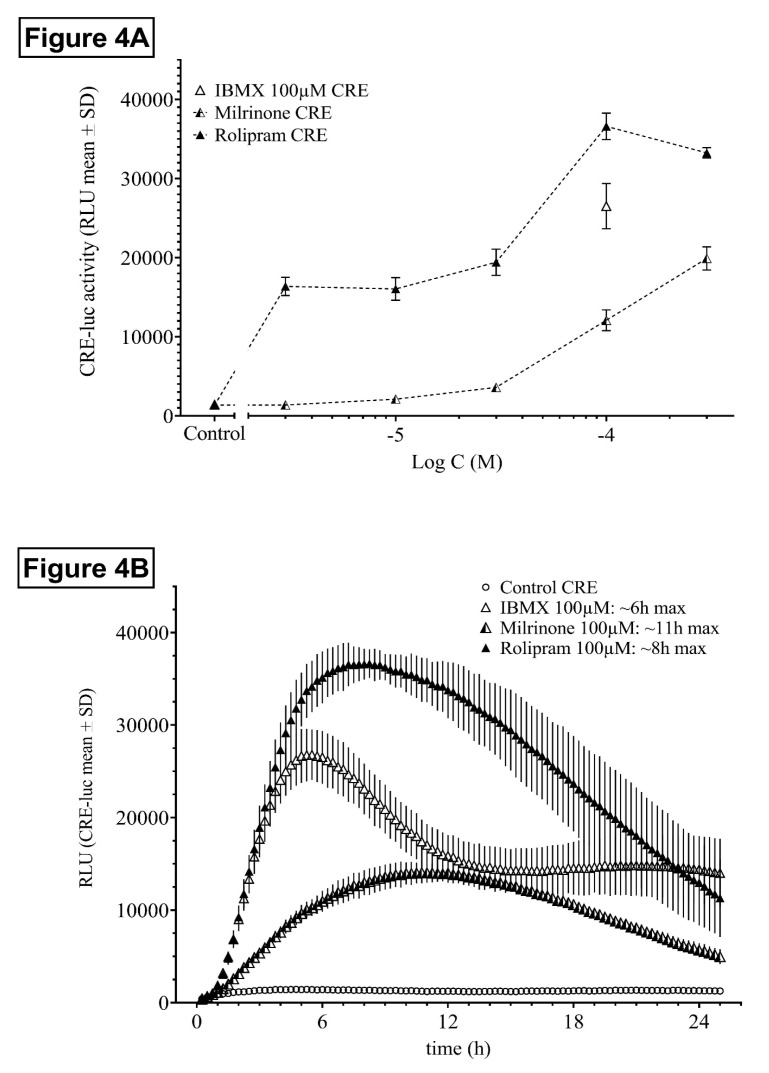
Dose- and time-dependent activity of the PDE-selective inhibitors rolipram and milrinone. (**A**) Dose-dependency of HT22CREluc activity for the PDE4-selective inhibitor rolipram and the PDE3-selective inhibitor milrinone in comparison to IBMX and untreated controls. Mean values with standard deviation (mean ± SD) of four repetitions (*N* = 4) are shown. (**B**) Time-resolved curves for all three inhibitors at the same dose (100 µM) in comparison to control (untreated) levels (mean ± SD; *N* = 4). Note the late onset and maximum activity of milrinone (~11 h) in comparison to rolipram (~8 h) and IBMX (~6 h) and the different offset of activity rates. All data are from the Centro luminometer.

**Figure 5 ijms-21-08658-f005:**
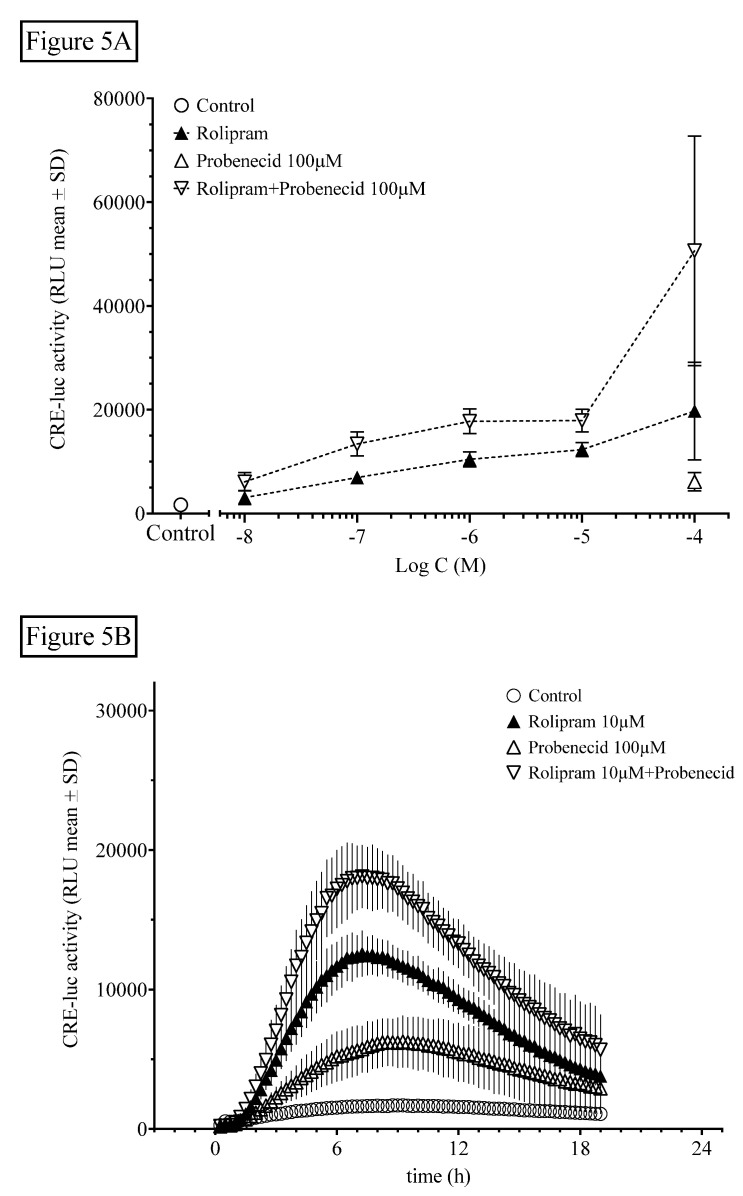
The organic anion-exchanger inhibitor probenecid further elevates rolipram-induced HT22CREluc activity. (A) HT22CREluc activity, elevated dose-dependently by the PDE selective inhibitor rolipram, was further enhanced by co-application of a single dose of the OAE inhibitor probenecid (100 µM). Mean values with standard deviation (mean ± SD) of four repetitions (N = 4) are shown. (B) Time-resolved curve with probenecid (100 µM), rolipram (10 µM), and the combination of both in comparison to control (untreated) levels (mean ± SD; N = 4). Note the later onset and maximum of HT22CREluc activity induced by probenecid application (~9 h) in comparison to rolipram (~7 h). All data are from the Centro luminometer.

**Figure 6 ijms-21-08658-f006:**
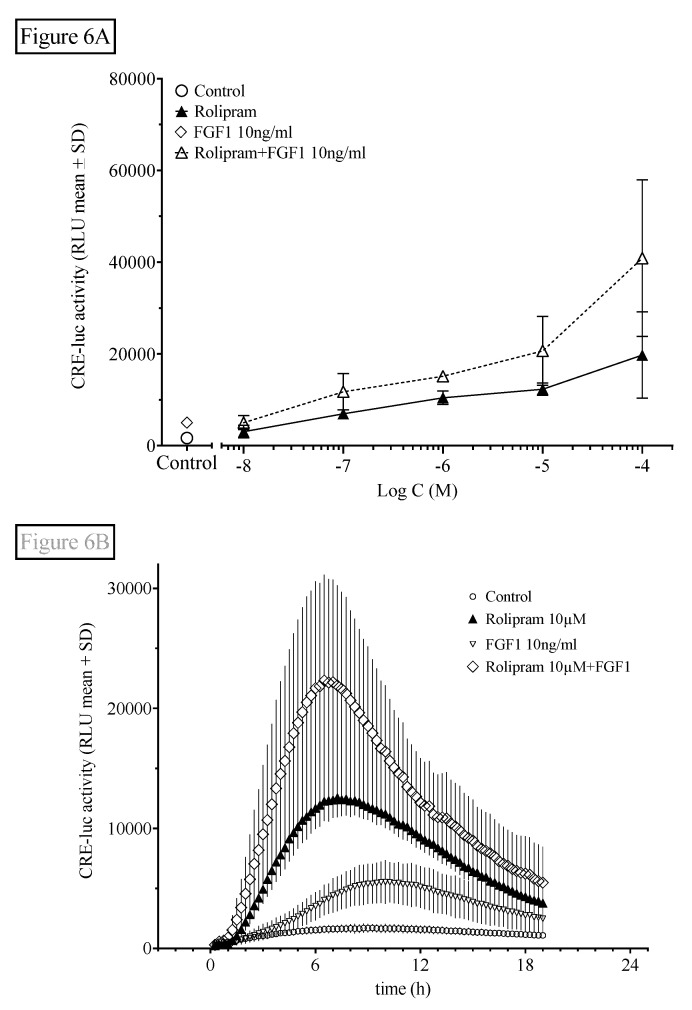
Fibroblast growth factor 1 (FGF1) application further elevates rolipram-induced HT22CREluc activity. (**A**) HT22CREluc activity, elevated by the PDE4-selective inhibitor rolipram, was further enhanced by co-application of FGF1 (10 ng/mL). Mean values with standard error (mean ± SD) of four repetitions (*N* = 4) are shown. (**B)** Time-resolved curve with FGF1 (10 ng/mL), rolipram (10 µM), and the combination of both in comparison to control (untreated) levels (mean ± SD; *N* = 4). Note the later onset and maximum of HT22CREluc activity induced by FGF1 (~9 h) in comparison to rolipram (~7 h). All data are from the Centro luminometer.

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
