# Peer review of "Influence of Phosphodiesterase Inhibition on CRE- and EGR1-Dependent Transcription in a Mouse Hippocampal Cell Line"

_ijms, 2020, doi:10.3390/ijms21228658_

Round 1

Reviewer 1 Report

This article described the dynamic of luminescence activity produced by reporter cell lines under the simulations of phosphodiesterase inhibitors with or without other reagents. The author showed dose- and time-dependent results and proposed potential applications of this system.

There are several concerns about this manuscript.

  1. Caffeine was mentioned in the abstract but not in the text.
  2. The values of RLU varied between figures; besides, the expression of these values also varied (mean with standard deviation or standard error; mean ± SD, mean ± SEM, mean + SD).
  3. Figure 1B and figure 2B are very similar, the last sentence in figure 2B was repeated.
  4. The content of figure legend 3 from line 130 might be in the text.
  5. Results about the co-treatment of OAE inhibitor and FGF1 agonist are interesting, the author should further elaborate these findings and explain “more than additive (synergistic) to rolipram (Figure5B)”.

Author Response

There are several concerns about this manuscript.

  1. Caffeine was mentioned in the abstract but not in the text.

Caffein is no longer mentioned in the text. However, I have done most experiments that were performed with IBMX also with Caffein and may add this as a supplement if the reviewer wishes so.

  1. The values of RLU varied between figures; besides, the expression of these values also varied (mean with standard deviation or standard error; mean ± SD, mean ± SEM, mean + SD).

Experiments in 1B and 2B are repetitions and were run on two different luminometers with different sensitivities (and different ages) as stated in the M&M section. Thus, the data in 1B and 2B show that the experimental outcomes are very similar even performed in different readers. RLU in the same reader does also change minimally from experiment to experiment presumably depending on cell passage number and other smaller influences. I could have normalized the RLU values but did not do so because normalization did not influence the findings. Therefore, I decided to show original data.

All graphs have been changed to “mean ± SD”.

  1. Figure 1B and figure 2B are very similar, the last sentence in figure 2B was repeated.

Figure 1 is data from the Luminometer “Centro” (“Berthold”, Germany)  and figure 2B is from the luminometer “Lumistar” ( BMG, Germany). As stated above experiments in 1B and 2B are repetitions and were run on two different luminometers with different sensitivities (and different ages) as stated in the M&M section. Thus, the data in 1B and 2B show that the experimental outcomes are very similar even performed in different readers. Figure 1b and 2B has been corrected accordingly.

  1. The content of figure legend 3 from line 130 might be in the text.

The content of figure legend 3 from line 130 on has been moved to the text.

  1. Results about the co-treatment of OAE inhibitor and FGF1 agonist are interesting, the author should further elaborate these findings and explain “more than additive (synergistic) to rolipram (Figure5B)”.

The findings on co-treatment of rolipram with probenecid and FGF1 have been elaborated further.

Reviewer 2 Report

The study developed by E. Maronde was aimed to explore the influence of phosphodiesterase inhibition on CRE and EGR1-dependent transcription in two reporter cell lines.

The subject is important due to the inhibitors analysed in this study are currently being used in clinical treatments, and therefore is essential to detect if they can produce side-effects. But, although the main goal of this study is well-addressed, resulting data need to be implemented with additional information.

  1. Regarding the writing style of the manuscript, abstract and introduction should be re-written in order to make them more understandable. It would be very helpful for readers to explain firstly what factors (AC activation and degradation through PDE) are controlling the levels of cAMP and describe its main effectors. At the present form, abstract and introduction are confusing. In addition, the reason to analyse OAE and FGF-1 is not clear in the introduction section. This information should be reasoned and complemented.

  1. Results 2.1.1. Figure 1B is not explained along the main text and results related with the dose of luciferin (from lines 88-91) are not shown.

  1. Results 2.1.2. “as expected from previous observations”.  I consider that “these observations” is the result that the author has to demonstrate with their experiments, so this sentence is unnecessary.

On the other hand, why does the author use a logarithmic scale in order to indicate the concentrations? This results confusing and unnecesary. It should be better use molarity (or micromolarity) units. (Fig 2,4,5,6)

  1. Results 2.1.3. By analysis with Rp-8Br-cAMPS, it seems likely that there is another effector different form PKA. Could the author indicate which effector can be involved?
  2. Results 2.1.4. How is the effect of milrinone and roipram in HT22EGFR1luc cell line? The author should indicate the reason for choosing only one cell line.
  3. Results 2.1.5. In Fig5B, the dosis of rolipram is 10 uM or 100 uM?
  4. Results 2.1.6. In Fig5B, the dosis of rolipram is 10 uM or 100 uM? From the graph, it seems that there is a synergestic effect of FGF1 on rolipram. Why does the author deny it?
  5. Discussion: In this study, two PDE inhibitors for PD3 and PD4 are employed but it is mentioned that there are eleven isozyme families. Are available other inhibitors for these different isozymes? Are they used in clinical treatments?
  6. The manuscript should have included a scheme indicating the molecular pathways analyzed in this present study, and also the inhibitors and agonist or antagonists molecules.
  7. Results from Section 4.3 is not shown in the main text.

Minor points:

Line 16: PDE abbreviation is not defined.

Line 18: caffeine is not employed in the present study

Line 48: “this manuscript” could not be a reference

Format of Figure 2A and 2B are different (Y axis) and the same for Figure 4

Figure 2 legend: Lines 105-106 are repeated.

Figure 3 legend: IBMX effect is notmediated exclusively by PKA because the addition of Rp-8Br-cAMPS to IBMX does not abolish luc activity.

Figure 6. Symbols for FGF1 alone and Rolipram+FGF1 are very similar. They should be modified.

Line 257: format for temperature is different in line 257,264,282

Line 258 “Living Cells” should be Living cells.

Author Response

Reviewer 2

The study developed by E. Maronde was aimed to explore the influence of phosphodiesterase inhibition on CRE and EGR1-dependent transcription in two reporter cell lines.

The subject is important due to the inhibitors analyzed in this study are currently being used in clinical treatments, and therefore is essential to detect if they can produce side-effects. But, although the main goal of this study is well-addressed, resulting data need to be implemented with additional information.

  1. Regarding the writing style of the manuscript, abstract and introduction should be re-written in order to make them more understandable. It would be very helpful for readers to explain firstly what factors (AC activation and degradation through PDE) are controlling the levels of cAMP and describe its main effectors. At the present form, abstract and introduction are confusing. In addition, the reason to analyse OAE and FGF-1 is not clear in the introduction section. This information should be reasoned and complemented.

Abstract and introduction were re-arranged according to the suggestions of reviewer 2. The reason to investigate OAE probenecid and FGF1 are elaborated now.

  1. Results 2.1.1. Figure 1B is not explained along the main text and results related with the dose of luciferin (from lines 88-91) are not shown.

Figure 1B is now explained in the text and a supplementary figure to explain the choice of the luciferin concentration is added.

  1. Results 2.1.2. “as expected from previous observations”.  I consider that “these observations” is the result that the author has to demonstrate with their experiments, so this sentence is unnecessary.

The phrase “as expected from previous observations” has been removed.

On the other hand, why does the author use a logarithmic scale in order to indicate the concentrations? This result is confusing and unnecessary. It should be better use molarity (or micromolarity) units. (Fig 2,4,5,6)

I use logarithmic scale because data span over many orders of magnitude, -3 means millimolar (mM), -6 micromolar (µM), -9 nanomolar (nM). I would be very happy to keep the graphs as they are and would kindly like to ask for the allowance to keep them like that.

  1. Results 2.1.3. By analysis with Rp-8Br-cAMPS, it seems likely that there is another effector different form PKA. Could the author indicate which effector can be involved?

Yes, Poppe et al., 2008 show that Rp-8-Br-cAMPS inhibits PDE4 with a Ki of 29µM and also other PDEs. Since 1 use 500µM Rp-8-Br-cAMPS here this may well influence the results in the sense that complete inhibition may be impossible.

  1. Results 2.1.4. How is the effect of milrinone and rolipram in HT22EGFR1luc cell line? The author should indicate the reason for choosing only one cell line.

Only preliminary experiments have so far been done with the HT22EGR1luc line on rolipram and milrinone (both activate, but much weaker than IBMX) and I decided to first evaluate the interactions of rolipram with other actors and signaling pathways using the HT22CREluc line.

  1. Results 2.1.5. In Fig5B, the dosis of rolipram is 10 uM or 100 uM?

10µM. The reason is because I concluded from the dose-response curve of rolipram that 10 µM (or maybe even up to 100µM)  is the specific effect of rolipram on PDE4 and at higher dose rolipram presumably inhibits other PDEs as well thereby obscuring the combination effects with other probenecid and FGF1.

  1. Results 2.1.6. In Fig6B, the dosis of rolipram is 10 uM or 100 uM?

10µM as stated in the graph and in the legend.

From the graph, it seems that there is a synergistic effect of FGF1 on rolipram. Why does the author deny it?

I do not deny it, it is not significant at 100µM which is now also visible from the data shown as means + SD). In the latter display overlap of error bars means non-significance which is also confirmed (as before) by the statistical post-test.

  1. Discussion: In this study, two PDE inhibitors for PD3 and PD4 are employed but it is mentioned that there are eleven isozyme families. Are available other inhibitors for these different isozymes? Are they used in clinical treatments?

Indeed, there are many more clinically relevant selective PDE inhibitors, here I concentrated on those two for which previous work suggested a relevance for hippocampal tissue function.

  1. The manuscript should have included a scheme indicating the molecular pathways analyzed in this present study, and also the inhibitors and agonist or antagonists’ molecules.

A scheme has been made and is provided in the supplements.

  1. Results from Section 4.3 is not shown in the main text.

Results from section 4.3 are now included in the main text.

Minor points:

Line 16: PDE abbreviation is not defined.

PDE has been defined

Line 18: caffeine is not employed in the present study

Caffein has been taken out of the abstract. However, I have done the experiments shown for IBMX also for caffein and may include them in a supplement if the reviewer wishes so.

Line 48: “this manuscript” could not be a reference

“this manuscript” has been eliminated.

Format of Figure 2A and 2B are different (Y axis) and the same for Figure 4

Figure 1 is data from the Luminometer “Centro” of the company “Berthold” and figure 2 is from the luminometer “Lumistar” of the company BMG. Thus, Figure 2B is an independent repetition of the data shown in 1B. Figure 1B and 2B has been corrected accordingly.

Figure 2 legend: Lines 105-106 are repeated.

The repeated lines 105-106 have been removed.

Figure 3 legend: IBMX effect is not mediated exclusively by PKA because the addition of Rp-8Br-cAMPS to IBMX does not abolish luc activity.

I agree with reviewer #2. Figure 3 legend has been corrected.

Figure 6. Symbols for FGF1 alone and Rolipram+FGF1 are very similar. They should be modified.

Figure 6 symbols have been changed and Mean + SD is shown instead of SEM.

Line 257: format for temperature is different in line 257,264,282

Temperature format has been unified.

Line 258 “Living Cells” should be Living cells.

“Living Cells” has been changed to “Living cells”
